# Spatial risk modelling of highly pathogenic avian influenza in France: Fattening duck farm activity matters

Jean Artois[1,2], Timothée Vergne[3], Lisa Fourtune[3], Simon Dellicour[1,4], Axelle Scoizec[5], Sophie Le Bouquin[5], Jean-Luc Guérin[3], Mathilde C. Paul[3], Claire Guinat[3]*

1 Spatial Epidemiology Lab (SpELL), Université Libre de Bruxelles, Brussels, Belgium, 2 Unité Productions Végétales, Centre Wallon de Recherches Agronomiques (CRAW), Gembloux, Belgium, 3 Interactions Hôtes-Agents Pathogènes (IHAP), UMR 1225, INRAE, ENVT, Université de Toulouse, Toulouse, France, 4 Department of Microbiology, Immunology and Transplantation, Rega Institute, KU Leuven, Leuven, Belgium, 5 Agence Nationale de Sécurité Sanitaire de l'Alimentation, de l'Environnement et du Travail (Anses), Ploufragan, France

* claire.guinat@envt.fr

**Data Availability Statement:** Detailed data about the poultry farm densities and poultry trade used in this study are not publicly accessible due to sensitive farmer information and are owned by the

## Abstract

In this study, we present a comprehensive analysis of the key spatial risk factors and predictive risk maps for HPAI infection in France, with a focus on the 2016–17 and 2020–21 epidemic waves. Our findings indicate that the most explanatory spatial predictor variables were related to fattening duck movements prior to the epidemic, which should be considered as indicators of farm operational status, e.g., whether they are active or not. Moreover, we found that considering the operational status of duck houses in nearby municipalities is essential for accurately predicting the risk of future HPAI infection. Our results also show that the density of fattening duck houses could be used as a valuable alternative predictor of the spatial distribution of outbreaks per municipality, as this data is generally more readily available than data on movements between houses. Accurate data regarding poultry farm densities and movements is critical for developing accurate mathematical models of HPAI virus spread and for designing effective prevention and control strategies for HPAI. Finally, our study identifies the highest risk areas for HPAI infection in southwest and northwest France, which is valuable for informing national risk-based strategies and guiding increased surveillance efforts in these regions.

## Introduction

Since November 2014, Europe has experienced several waves of incursions of highly pathogenic avian influenza (HPAI) A(H5Nx) clade 2.3.4.4b virus [1, 2]. The 2020–21 wave was an unprecedented epidemic, with a total of 3,791 HPAI poultry farm outbreaks and wild bird cases in 31 European countries, outreaching those caused by the 2016–17 wave (2,781 in 29 countries) [3]. The 2021–22 wave has now become the largest epidemic wave, with 6,227 cases in 37 countries, as of September 2022 [4]. During the different waves, HPAI A(H5Nx) virus

Direction Générale de l'Alimentation, Ministère de l'Agriculture et de l'Alimentation (DGAl) of the Ministère de l'Agriculture et de la Souveraineté Alimentaire, France. Data requests can be made to the DGAl (https://agriculture.gouv.fr). Data about poultry outbreaks can be found on the Global Animal Disease Information System (https://empres-i.apps.fao.org) of the Food and Agriculture Organization of the United Nations (FAO). The R code about the spatial smoothing of variables is available online (10.5281/zenodo.13765228).

**Funding:** This study was performed in the framework of the Chair for Avian Biosecurity and Health, hosted by the National Veterinary College of Toulouse and funded by the Direction Générale de l'Alimentation, Ministère de l'Agriculture et de l'Alimentation, France. Jean Artois and Simon Dellicour have received funding from the European Union's Horizon 2020 research and innovation programme under grant agreement No 874850. The contents of this publication are the sole responsibility of the authors and don't necessarily reflect the views of the European Commission. Simon Dellicour also acknowledges support from the Fonds National de la Recherche Scientifique (F.R.S.-FNRS, Belgium; grant n°F.4515.22) and from the Research Foundation - Flanders (Fonds voor Wetenschappelijk Onderzoek-Vlaanderen, FWO, Belgium; grant n°G098321N). The funders had no role in study design, data collection and analysis, decision to publish, or preparation of the manuscript.

**Competing interests:** The authors declare no conflict of interest.

rapidly evolved and reassorted, leading to the co-circulation of several subtypes, with H5N8 being predominant in 2016–17 and 2020–21, and H5N1 in 2021–22 [5–8]. The persistent occurrence of devastating HPAI A(H5Nx) epidemics in Europe over the last few years raises concerns about the capacity of applied biosecurity measures to prevent virus introduction and about our mitigation strategies after virus introduction.

France has been one of the most severely and repeatedly affected European countries by the successive epidemic waves [5, 9, 10]. In 2016–17 and 2020–21, most of the poultry farm outbreaks occurred in fattening duck flocks in southwestern France (Fig 1).

This was likely due to very high poultry farm densities but also to the particularities of this production type, including outdoor rearing practices and strong farm connectivity [9, 11–13]. Notably, the number of wild bird cases remained limited during these periods [10] (Fig 1). Several epidemiological and phylogenetic studies have shown that farm-to-farm transmission and human activities-mediated transmission were likely the main drivers of the spatial spread of the HPAI H5N8 virus during the 2016–17 and 2020–21 waves in France [11, 14–16], rather than continuous incursion events from a wild bird reservoir. However, the higher number and wider range of infected wild bird species reported during the 2021–22 wave may support the hypothesis that the H5N1 virus has spread in the wild bird population more extensively than ever before, increasing the opportunity for the virus to spill over into poultry and making the epidemic particularly difficult to contain [17].

By comparison with other types of poultry production, the fattening duck production is highly segmented, with three main production stages [18]: rearing (~3 weeks), breeding (~9 weeks) and force-feeding (~12 days). The breeding stage can also be further divided into starting, growing and pre-force-feeding periods. Each stage/period is associated with specific farming practices, equipment and resources and often involves multiple and different farms at different geographical locations [15, 19]. As a result, fattening duck farms are highly connected, through the frequent movements of live ducks, vehicles, equipment or humans, which can represent opportunities for virus farm-to-farm transmission. For instance, during the 2016–17 wave, some of the earliest poultry farm outbreaks were likely associated with the movements of live ducks and vehicles, prior to the implementation of movements' restrictions [12, 13]. In addition, inadequate farm access control systems and management of vehicles and visitors entering or leaving the farms were identified as the major biosecurity breaches contributing to the risk of HPAI introduction [11]. During the different epidemic waves, it is also possible that airborne transmission of the HPAI virus has contributed to the disease transmission over short distances [20–22].

Based on data collected during the 2016–17 wave, Guinat et al. [15] applied a boosted regression trees methodology to identify spatial risk factors for HPAI infection and generate associated predictive risk maps. The outcomes of this study significantly helped provide actionable advice and guidance to the French authorities on how to mitigate the risk of HPAI infection in France during the following epidemic waves [23, 24]. For instance, the trade-related transport of fattening ducks was identified as one of the major factors influencing the spatial distribution of outbreaks during the 2016–17 wave. This triggered French authorities and farmer organizations to reinforce biosecurity along the transport of ducks. This included improved cleaning and disinfection procedures of cages and trade-related vehicles, coverage of vehicles with nets during high-risk periods and the use of different sets of cages between the different production stages [25]. The predictive risk maps were also essential to define the high-risk areas for HPAI infection, where biosecurity measures were reinforced during the 2021–22 wave [23]. This resulted in a list of 539 municipalities identified as high-risk areas for HPAI infection.

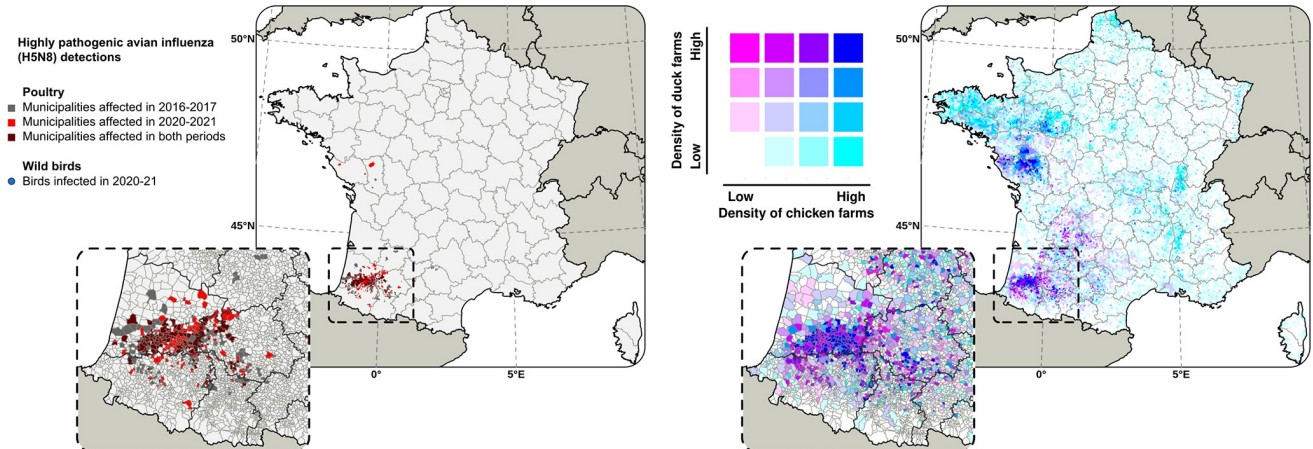

**Fig 1. Poultry farm outbreaks and densities.** Geographical distribution of (i) the French municipalities with at least one highly pathogenic avian influenza H5N8 poultry farm outbreak reported during the 2016–17 or the 2020–21 waves (left panel), and (ii) poultry farm densities, with colour intensity proportional to the number of chicken farms per km$^2$ (purple) and the number of duck farms per km$^2$ (cyan). Data were obtained from the DGAl (Direction Générale de l′Alimentation of the French Ministry of Agriculture) and CIFOG (Comité Interprofessionnel des Palmipèdes à Foie Gras). Shapefiles used to create maps are based on administrative boundaries available in the public domain (CC BY 4.0).

Despite recent advances in our understanding of the spread of HPAI in France, further research is necessary to gain a deeper insight into this complex mechanism. To this end, we utilized the opportunity presented by the 2016–17 and 2020–21 waves to update and compare the most important spatial risk factors and predictive risk maps for HPAI infection in France. Our approach leveraged scale-optimization techniques and generalized linear mixed models to address spatial dependencies in the data. The results of our analysis inform the development of national risk-based strategies by highlighting the regions in France that require increased surveillance efforts. Additionally, our analytical approach can be applied in other regions to assess the appropriate spatial scale for the analysis of epidemiological data and communication of HPAI risk.

## Methods

### Data collection

**Outbreak data.**   Data related to all confirmed HPAI H5N8 poultry outbreaks during the 2020–21 wave (04/11/2020–20/03/2021, $n = 469$) were obtained from the French Ministry of Agriculture (DGAL). A confirmed outbreak was defined as the detection of at least one laboratory-confirmed HPAI H5N8 infected bird (by virus isolation or PCR) in a poultry house. To fit the scale at which prevention and mitigation strategies are implemented, outbreak data were aggregated at the municipality level and summarized as the number of infected HPAI H5N8 poultry houses per municipality.

**Poultry data.**   Six variables, related to poultry house densities were considered for their roles in the spatial distribution of HPAI H5N8 infected poultry houses per municipality (Table 1): the number of poultry (chicken and duck) houses, the density of chicken houses per km$^2$, the density of duck houses per km$^2$, the density of fattening duck (all stages) houses per km$^2$, the density of fattening duck (breeding stage) houses per km$^2$ and the density of fattening duck (force-feeding stage) houses per km$^2$.

The list of the different types of poultry houses per municipality was obtained from the DGAl (Direction Générale de l′Alimentation of the French Ministry of Agriculture), from which the respective densities of poultry per km$^2$ were computed.

**Table 1. List of predictor variables.** The final spatial predictor variables were calculated using different levels of municipality aggregation (see *Scale-optimized risk models* section). Sources: DGAl (Direction Générale de l'Alimentation of the French Ministry of Agriculture) and CIFOG (Comité Interprofessionnel des Palmipèdes à Foie Gras).

| Data type | Variable |
|---|---|
| **Poultry population data** | Number of poultry (chicken and duck) houses |
| | Density of chicken houses ($/km^2$) |
| | Density of duck houses ($/km^2$) |
| | Density of fattening duck (all stages) houses ($/km^2$) |
| | Density of fattening duck (breeding stage) houses (/km2) |
| | Density of fattening duck (force-feeding stage) houses ($/km^2$) |
| **Live duck movement data** | Density of outgoing fattening duck flock movements (breeding to force-feeding stage) ($/km^2$) |
| | Density of incoming fattening duck flock movements (breeding to force-feeding stage) ($/km^2$) |
| | Number of municipalities sending fattening duck flocks to the given municipality (breeding to force-feeding stage) |

Given the variables related to the density of poultry houses may not encompass information regarding their operational status, e.g., whether they are active or not, we included the density of pre-epidemic duck flock movements (e.g., before the implementation of movement ban restrictions during the epidemic) to gain more precise information of farm activity. Three variables, related to duck flock movement densities, were thus considered per municipality (Table 1): the density of outgoing fattening duck flock movements (breeding to the force-feeding stage) per $km^2$, the density of incoming fattening duck flock movements (breeding to the force-feeding stage) per $km^2$ and the number of municipalities sending fattening duck flocks to the targeted municipality (breeding to the force-feeding stage). The list of movements of outgoing and incoming fattening duck flocks between municipalities was obtained from the CIFOG (Comité Interprofessionnel des Palmipèdes à Foie Gras) over the period prior to the 2020–21 epidemic wave (16/06/2020–03/11/2020). The dataset contained 29,005 observations and included information about the location of fattening duck houses, the flock production stage (rearing, breeding or force-feeding), the flock destination (farm or slaughterhouse) and the movement date. We exclusively focused on movements occurring between breeding and force-feeding houses, because they represented the most complete dataset and the majority of the total number of movements, totaling 12,507 observations. Consequently, we excluded from our analysis movements involving breeding houses, rearing houses, and slaughterhouses. Also, only duck movements were considered, given that the majority of outbreaks were reported within this species and because of the higher frequency of inter-stage movements inherent to this specific production system.

Noteworthily, no variables related to wild bird populations or associated factors such as climate or landscape features were included in this study. This was based on several key considerations: during that specific wave, infected wild birds were identified as the primary source of virus introduction into the country, while their role in the diffusion of the virus between poultry farms was found negligible [5, 26]. In addition, previous research during the 2016–17 wave, which exhibited similarities to the 2020–21 wave, revealed that human population density had a negligible contribution to HPAI H5N8 infection [15].

## Risk mapping

**Study area.** We carried out an analysis at a local spatial scale that only included municipalities in southwestern France, and in particular those located in one of the four most affected

departments (administrative units with a median surface of 6,000 km$^2$) (namely Landes, Gers, Pyrénées-Atlantiques, Hautes-Pyrénées), to identify the fine-scale spatial predictor variables for HPAI H5N8 infection in this highly affected region. The spatial extent of this local-scale analysis is shown in Fig 2.

We produced predictive risk maps for HPAI infection for the western half of the country based on the statistical model trained with the local-extent analysis. The area used for predictions is shown in Fig 2. Municipalities categorised as high-risk areas for HPAI infection during the 2016–17 wave [15] are shown in Fig 2.

**Scale-optimized risk models.** Although high densities of poultry houses at the municipality level are reported throughout the western part of the country, the number of adjacent municipalities producing fattening ducks is particularly high in southwestern France. Therefore, we hypothesised that the degree to which municipalities producing poultry are spatially clustered is an important factor to be considered in assessing the risk of HPAI presence at the municipality level. In other words, we hypothesised that the risk of infection in a given municipality is dependent on the density of poultry houses in the wider area, i.e. not only in the given

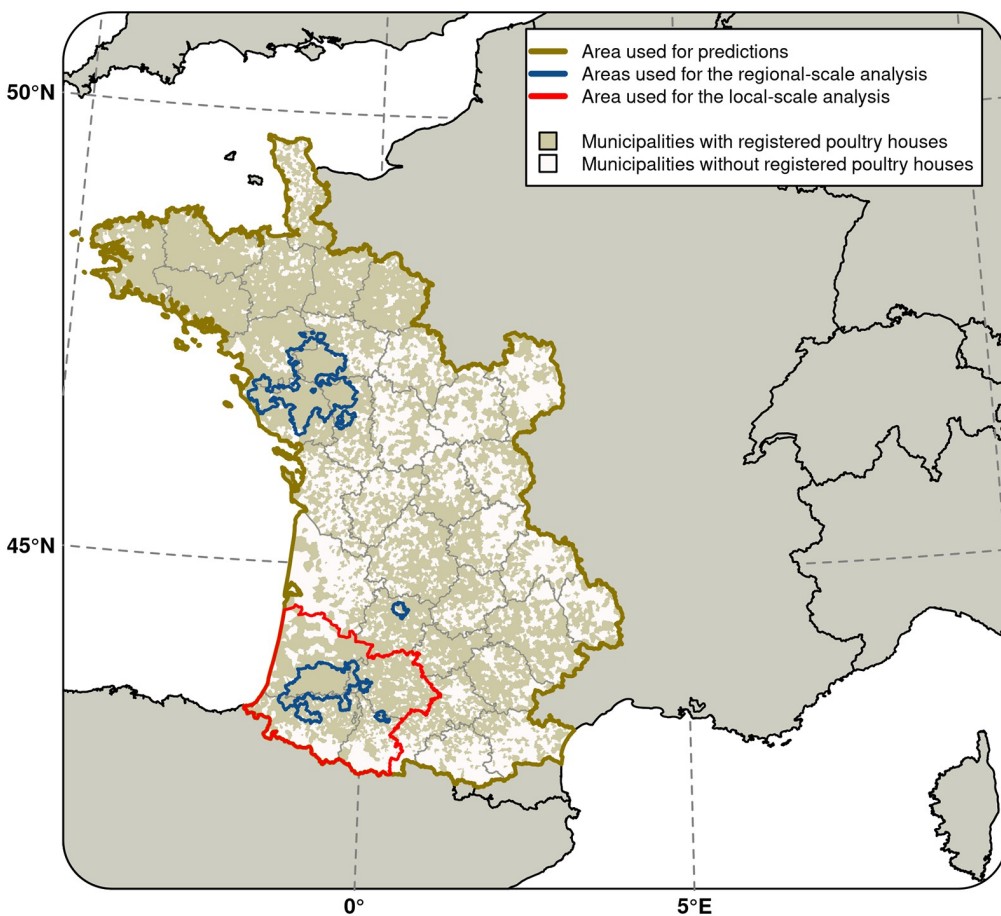

**Fig 2. Study areas.** The area used for the model predictions is depicted by the dark yellow contour. The areas used for the model training are depicted by the red line. The 539 municipalities identified as the high-risk areas for HPAI H5N8 infection during the 2016–17 wave [15] are depicted by the blue line. Only municipalities with poultry houses were included in the 2020–21 analysis and are depicted in beige, while municipalities without poultry houses were not included and are depicted in white. R software version 4.2.0 (https://cran.r-project.org/) was used to produce the figure. Shapefiles used to create maps are based on administrative boundaries available in the public domain (CC BY 4.0).

municipality but also in the adjacent municipalities. To account for this, two approaches were used: (i) First, the spatial predictor variables were smoothed based on five different levels (S1 Fig): level 0 included the given municipality, level 1 included municipalities with contiguous boundaries with the given municipality, level 2 included the list of the neighbouring municipalities from level 1, in addition to municipalities with contiguous boundaries with the neighbouring municipalities from level 1. This process was repeated four times in order to obtain four sets of neighbouring municipalities, which were then used to calculate the spatial predictor variables used for the modelling part. Maps of spatial predictor variables are available in S1 File. (ii) Second, spatial generalised linear mixed models (GLMM) were used to quantify the effect of the spatial predictor variables on the proportion of HPAI H5N8 infected poultry houses per municipality and to generate associated predictive risk maps. Random effects with a Besag York Mollie (BYM) conditional autoregressive (CAR) prior [27, 28] were used to take into account spatial auto-correlation and produce a spatially smooth map of the proportion of HPAI H5N8 infected poultry houses per municipality. A BYM model includes both an intrinsic CAR model and an independent and identically distributed Gaussian random effect model (iid) to take into account spatially and non-spatially structured error terms.

**Setting parameters of risk models.** The models were developed using a binomial distribution, with the number of infected poultry houses per municipality as the number of successes and the total number of poultry houses per municipality as the number of trials. The models were fitted using the integrated nested Laplace approximation (INLA) [29] specifying parameters of prior distributions to log-Gamma (1, 0.00005) for the spatial unstructured and structured random effects which is the default specification of these parameters on R-INLA. In order to limit over-parametrization of the GLMM, the untransformed spatial predictor variables were selected based on a forward selection procedure using the widely applicable information criterion (WAIC) [30] to compare the models. The models with the lowest WAIC values were considered as the best trade-offs between a reduction of the model errors and the over-parametrization of models. WAIC differences greater than or equal to 5 were only considered in the model comparison process to focus on the main effects influencing the spatial distribution of HPAI H5N8 outbreaks. We performed a forward model selection procedure using the iid models. The final iid model was then compared to a BYM model based on the same untransformed spatial predictor variables selected during the forward model selection procedure. R software version 4.2.0 [31] and the sf [32, 33], spdep [34] and R-INLA [35] packages were used for spatial data manipulation, analyses and to produce the figures and maps of this manuscript. The R code about the spatial smoothing of variables is available online (10. 5281/zenodo.13765228).

**Validation of risk models.** To evaluate the final model for its capacity to discriminate between the presence and the absence of HPAI H5N8 infections at the municipality level, the predicted proportion of HPAI H5N8 infected poultry houses per municipality was converted into a probability of having at least one outbreak in the municipality. The probability of having at least one outbreak $P(X > 0)$ was estimated with a binomial distribution as follows: $P(X > 0) = 1 - (1-p)^n$, where n is the number of poultry houses in the municipality, and p is the proportion of HPAI H5N8 infected poultry houses predicted by the final binomial model. Then, the discriminatory capacity of the final binomial model was assessed using the area under the ROC curve [36, 37] over the 2016–17, 2020–21, and 2021–22 epidemic waves. Aggregated data related to the presence and the absence of HPAI H5N8 infections at the municipality level for the 2016–17 and 2021–22 waves were obtained from the French Ministry of Agriculture (DGAL). Ultimately, the discriminatory capacity of the final binomial model was compared to the Guinat et al. [15] boosted regression trees local-extent model.

Additional information regarding the ethical, cultural, and scientific considerations specific to inclusivity in global research is included in the Supporting Information (S1 Checklist).

## Results

Two spatial predictor variables were selected in the final model (Table 2): the density of incoming duck movements from breeding to force-feeding houses at level 3 (which includes three spatial rings of municipalities surrounding the given municipality) and the density of outgoing duck movements from breeding to force-feeding houses at level 0 (which only includes the given municipality).

The Pearson correlation coefficient between the selected variables was equal to 0.31. Both variables were positively associated with the proportion of HPAI H5N8-infected poultry houses per municipality (Table 3).

We compared this final model to an alternative model (Table 2) that was considered as equivalent by replacing the density of incoming duck movements from breeding to force-feeding houses at level 3 with the density of force-feeding houses at level 3 and by replacing the density of outgoing duck movements from breeding to the force-feeding stage at level 0 by the density of breeding houses at level 0. This alternative model showed similar but slightly higher WAIC scores compared to the final model, with both variables also positively associated with the proportion of HPAI H5N8 infected poultry houses per municipality (Table 3). In the final and alternative models, the iid model shows very similar WAIC scores when compared to the BYM model which considers spatial auto-correlation effects. A more complete summary of the final model fits is available in S1 Table.

Fig 3 shows the predictive maps of the proportion of HPAI H5N8 infected poultry houses per municipality and the probability of having at least one HPAI H5N8 poultry outbreak per commune.

One major high-risk area was identified in southwestern France, in municipalities located along the border between Landes, Gers, Hautes-Pyrénées and Pyrénées-Atlantiques departments. The risk slightly increased in Lot, Lot-et-Garonne, and Dordogne, as well as in northwest France, in municipalities of Deux-Sevres, Loire-Atlantique, Maine-et-Loire, Mayenne, Sarthe and Vendée. S2 Fig shows that the final and alternative models generated similar high-risk areas for HPAI infection.

**Table 2. Forward model selection.** The widely applicable information criterion (WAIC) score differences between models are also provided. BYM: Besag York Mollie conditional autoregressive model; iid: Independent and identically distributed Gaussian random effect model.

| Model | | WAIC | WAIC difference |
|---|---|---|---|
| **Final model** | Intercept-only model (iid) | 1285.55 | - |
| | Density of incoming fattening duck flock movements ($/km^2$) at level 3 (iid) | 1203.43 | 82.12 |
| | Density of incoming fattening duck flock movements ($/km^2$) at level 3 + Density of outgoing fattening duck flock movements ($/km^2$) at level 0 (iid) | 1193.98 | 9.45 |
| | Density of incoming fattening duck flock movements ($/km^2$) at level 3 + Density of outgoing fattening duck flock movements ($/km^2$) at level 0 (BYM) | 1194.03 | -0.05 |
| **Alternative model** | Intercept-only model (iid) | 1285.55 | - |
| | Density of fattening duck (force-feeding stage) houses ($/km^2$) at level 3 (iid) | 1213.93 | 71.63 |
| | Density of fattening duck (force-feeding stage) houses ($/km^2$) at level 3 + Density of fattening duck (breeding stage) houses ($/km^2$) at level 0 (iid) | 1209.07 | 4.85 |
| | Density of fattening duck (force-feeding stage) houses ($/km^2$) at level 3 + Density of fattening duck (breeding stage) houses ($/km^2$) at level 0 (BYM) | 1210.61 | -1.54 |

**Table 3. Summary statistics for the final model and the alternative model.** Median and the lower and upper limits of 95% credible intervals of the posterior marginal distribution of variables' coefficients for the final model and the alternative model.

| Model | Variable | Median | 95% credible intervals |
|---|---|---|---|
| **Final model** | (intercept) | -4.49 | [-4.78; -4.21] |
| | Density of incoming fattening duck flock movements (/km$^2$) at level 3 | 3.46 | [2.80; 4.15] |
| | Density of outgoing fattening duck flock movements (/km$^2$) at level 0 | 0.27 | [0.12; 0.42] |
| **Alternative model** | (intercept) | -4.37 | [-4.65; -4.10] |
| | Density of fattening duck (force-feeding stage) houses (/km$^2$) at level 3 | 4.00 | [3.01; 5.05] |
| | Density of fattening duck (breeding stage) houses (/km$^2$) at level 0 | 0.15 | [0.00; 0.31] |

The probabilities of having at least one HPAI H5N8 poultry outbreak per municipality, as predicted by the final and alternative models, showed very good discriminatory accuracy, with AUC values of 0.942 and 0.937, respectively, for the 2020–21 wave (Table 4).

The final model performs better than the alternative model and the 2016–17 model [15] in predicting the presence of HPAI infections during the 2021–22 waves (0.844 compared to 0.830 and 0.825, respectively). However, for the 2021–22 wave, the discriminatory capacities of both the final and alternative models' predictions were lower (0.844 and 0.830), as compared to the 2016–17 (0.880 and 0.872) and 2020–21 waves (0.942 and 0.937). This could be attributed to the greater similarity in the list of infected municipalities between 2016–17 and 2020–21 waves, in contrast to the 2021–22 wave (Fig 1) [9].

## Discussion

This study expands our understanding of the key spatial risk factors for HPAI infection in France. The densities of outgoing and incoming duck movements from breeding to force-feeding houses during the pre-epidemic wave were selected as the most explanatory spatial

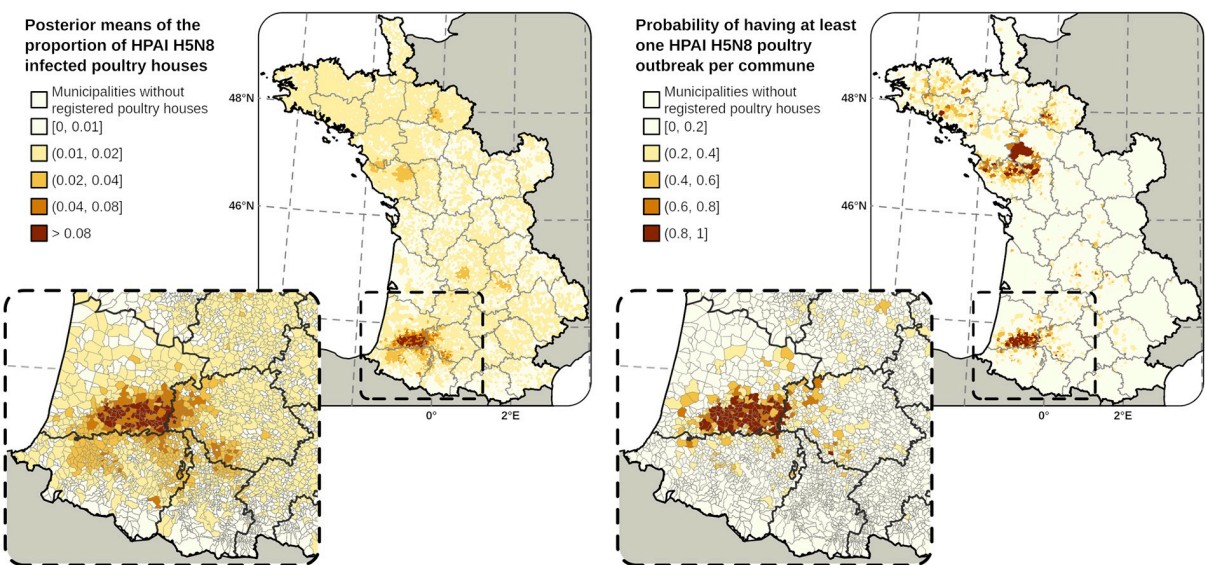

**Fig 3. Predictive maps.** Posterior means of the proportion of HPAI H5N8 infected poultry houses per municipality predicted by the final model (left panel). Probability of having at least one HPAI H5N8 poultry outbreak per commune (right panel). Shapefiles used to create maps are based on administrative boundaries available in the public domain (CC BY 4.0).

**Table 4. Discriminatory capacity of the different models.** The final models, the alternative model and the 2016–17 model [15] were assessed for different epidemic waves of HPAI H5N8 in France, as measured by the area under the ROC curve.

| | | Observed values | | |
|---|---|---|---|---|
| | | 2016–17 | 2020–21 | 2021–22 |
| **Predicted values** | **2016–17 model** | 0.887 | 0.892 | 0.825 |
| | **2020–21 final model** | 0.880 | 0.942 | 0.844 |
| | **2020–21 alternative model** | 0.872 | 0.937 | 0.830 |

predictor variables of the proportion of HPAI H5N8 infected poultry houses per municipality. The role of trade-related transport of fattening ducks in the occurrence of HPAI H5N8 outbreaks had already been identified during the 2016–17 epidemic wave (7), leading to reinforced biosecurity measures along the transport of ducks. However, our study only considered pre-epidemic duck flock movements, occurring prior to the implementation of movement ban restrictions during the epidemic, and should not be considered as a direct cause of outbreak occurrence but rather as indicators of farm operational status, e.g., whether they are active or not. This underscores the relationship between poultry house activity and the proportion of infected poultry houses, indicating that greater farm activity corresponds to a higher likelihood of future outbreak occurrence. Consequently, implementing measures to reduce poultry house activity or having poultry houses that operate at different periods across the year holds promise for more effective disease control strategies, as seconded by a recent modelling study [38].

Given the challenges associated with the collection of movement data, we also explored the possibility of using the density of breeding and force-feeding houses as potential proxies for the density of outgoing and incoming movements, respectively. To this end, the final model was compared with an alternative equivalent model based on the density of force-feeding houses at level 3 and the density of breeding houses at level 0. The results indicated that the alternative model produced similar but slightly higher WAIC and lower AUC scores compared to the final model. Thus, our results suggest that the density of fattening duck houses could be used as a valuable predictor of the spatial distribution of outbreaks per municipality. This is particularly relevant as data on the density of fattening duck houses is generally more readily available and easier to collect than data on movements between houses. However, the difference in prediction accuracy shows that the accuracy of housing density may be less precise as a reflection of operational status, e.g., whether houses are active or not, compared to the direct indicator provided by the existence of movements of fattening ducks from/to these houses. This also underscores the need to improve data precision and update for poultry house density, accounting for changes such as production cessation or periods during which those houses are full or empty with ducks, for future risk assessment studies.

Our variables were defined at different levels, each representing rings of municipalities expanding in size around a central municipality. This approach was used to account for the fact that while the municipality was the unit of interest, poultry farms are often part of extensive contact networks that extend beyond municipality borders including animal introduction networks (duck trade), transit networks (transportation vehicle transit), and farm personnel networks [13, 39–41]. Our study found that the scale-optimized density of incoming movements or of houses at level 3 was the most significant spatial predictor variable of the spatial distribution of outbreaks per municipality, which on average corresponds to a catchment area of 643 km$^2$. Notably, models with and without spatio-temporal effects were quite similar in terms of predictive performance. This finding highlights the efficacy of our approach with variables defined at different ring levels to account for spatial auto-correlation. This also supports the hypothesis that considering the operational status of duck houses in nearby municipalities

is crucial in predicting the risk of future HPAI infection for a given municipality. The scale-optimization approach used in this study enables the final model's capacity to capture risk differences over a broader spatial area than the existing municipalities.

While the risk models used in this study were based on data obtained from a single epidemic wave (2020–21), the risk maps of the mean proportion of HPAI H5N8 infected poultry houses per municipality align with those produced during the 2016–17 wave [15], with the highest risk observed in southwestern France. Furthermore, the two predicted high-risk areas closely matched the spatial distribution of poultry farm outbreaks during the 2021–22 epidemic wave [9, 42].

During the 2020–21 wave under study, it was recognized that wild birds served as the primary source of virus introduction into the country, while their contribution to the diffusion of the virus between poultry farms was found to be negligible. This was supported by the very limited number of wild bird cases during that specific wave and by phylogenetic analyses providing strong evidence of between-farm transmission [5, 26]. This prior knowledge guided our selection of predictors, which did not include variables related to wild birds or associated factors such as climate or landscape features. However, it is worth noting that since the 2021–22 wave, the H5N1 virus has spread more extensively than ever before among wild bird populations [9, 43, 44], thus increasing the risk of virus spillover into domestic poultry. To test this hypothesis, spatial risk models for the subsequent waves should incorporate wild bird-related variables, such as the spatial distribution of wild bird cases or indices of wild bird habitat suitability, to comprehensively assess the dynamics of HPAI H5N1 virus transmission. In our explanatory work, we also considered various potential variables, including farm biosecurity levels or staff and vehicle movements. However, their incorporation into the model was challenging due to the unavailability or lack of detailed data. Future research should focus on generating and collecting such data to better understand their role in the risk of infection. Thus, we focused on variables with actionable implications, such as duck movements and housing density, as these provide practical insights that can directly inform targeted interventions for effective outbreak control. We assumed the perfect reporting of infected houses, meaning that we did not account for the possibility of having more infected houses than officially reported. While this could impact our findings, this is a reasonable assumption due to the severe clinical signs due to HPAI H5N8 infection making them easily detectable, past experiences of farmers with previous outbreaks resulting in swift reporting practices, and the provision of financial compensation which further encourages timely reporting.

The persistent occurrence of devastating HPAI epidemics in France in recent years highlights the crucial need for research into methods to secure fattening duck production on a long-term basis. This study underscores the epidemiological relevance of reducing fattening duck farm activity as a key intervention to limit the risk of HPAI in the French poultry sector, as suggested in a previous study [38]. Short-term measures to achieve this include increasing the downtime between production cycles or limiting poultry placements during high-risk periods. However, implementing such structural changes requires further research to evaluate their sociological and economic impacts and ensure the sustainability and feasibility of such proposed interventions. The ultimate goal is to strike a balance between reducing the risk of HPAI transmission and ensuring the viability of the fattening duck sector. Additionally, this study guides the allocation of resources for increased surveillance and intervention efforts in regions identified as high-risk areas.

## Supporting information

**S1 Fig. Smoothing of the spatial predictors.** Spatial representation of a given municipality and neighboring municipalities across four spatial scales, from the level 1 (olive green) to the

level 4 (dark blue). Shapefiles used to create maps are based on administrative boundaries available in the public domain (CC BY 4.0).
(TIF)

**S2 Fig. Comparison of high-risk areas obtained using different statistical models.** The final model results are represented on top of the figure and the alternative model results are in the bottom part of the figure. The posterior means of the proportion of HPAI H5N8 infected poultry houses per municipality are represented on the left panels and the probability of having at least one HPAI H5N8 poultry outbreak per commune are represented on the right panels. Shapefiles used to create maps are based on administrative boundaries available in the public domain (CC BY 4.0).
(TIF)

**S1 File. Spatial distribution of predictor variables.** Predictor variables included the number of poultry (chicken and duck) houses, the density of chicken houses per $km^2$, the density of duck houses per $km^2$, the density of fattening duck (all stages) houses per $km^2$, the density of fattening duck (breeding stage) houses per $km^2$ and the density of fattening duck (force-feeding stage) houses per $km^2$. Shapefiles used to create maps are based on administrative boundaries available in the public domain (CC BY 4.0).
(PDF)

**S1 Table. Summary of the final binomial iid models fits.**
(XLSX)

**S1 Checklist.**
(DOCX)

## Acknowledgments

The authors specifically acknowledge Guillaume Gerbier (DGAl) for useful comments on the manuscript. The authors acknowledge the French National Reference Laboratory for avian influenza at ANSES and its accredited local laboratories' network that performed all the analyses for cases' confirmation, and the DGAl services in charge of avian influenza surveillance, management and control.

## Author Contributions

**Conceptualization:** Jean Artois, Timothée Vergne, Mathilde C. Paul, Claire Guinat.

**Data curation:** Jean Artois, Lisa Fourtune.

**Formal analysis:** Jean Artois.

**Supervision:** Timothée Vergne, Simon Dellicour, Axelle Scoizec, Sophie Le Bouquin, Jean-Luc Guérin, Mathilde C. Paul.

**Validation:** Lisa Fourtune.

**Writing – original draft:** Jean Artois, Claire Guinat.

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
