## [Decision Letter · Decision Letter 0]

28 Aug 2024

PONE-D-24-25856Spatial risk modelling of highly pathogenic avian influenza in France: fattening duck farm activity mattersPLOS ONE

Dear Dr.  Guinat,  Thank you for submitting your manuscript to PLOS ONE. After careful consideration, we feel that it has merit but does not fully meet PLOS ONE’s publication criteria as it currently stands. Therefore, we invite you to submit a revised version of the manuscript that addresses the points raised during the review process. Minor concerns raised by the reviewer shall be addressed properly. Please submit your revised manuscript by Oct 12 2024 11:59PM. If you will need more time than this to complete your revisions, please reply to this message or contact the journal office at plosone@plos.org. When you ready to submit your revision, log on to https://www.editorialmanager.com/pone/ and select the 'Submissions Needing Revision' folder to locate your manuscript file.

Please include the following items when submitting your revised manuscript:A rebuttal letter that responds to each point raised by the academic editor and reviewer(s). You should upload this letter as a separate file labeled 'Response to Reviewers'.A marked-up copy of your manuscript that highlights changes made to the original version. You should upload this as a separate file labeled 'Revised Manuscript with Track Changes'.An unmarked version of your revised paper without tracked changes. You should upload this as a separate file labeled 'Manuscript'.If applicable, we recommend that you deposit your laboratory protocols in protocols.io to enhance the reproducibility of your results. Protocols.io assigns your protocol its own identifier (DOI) so that it can be cited independently in the future. For instructions see: https://journals.plos.org/plosone/s/submission-guidelines#loc-laboratory-protocols. Additionally, PLOS ONE offers an option for publishing peer-reviewed Lab Protocol articles, which describe protocols hosted on protocols.io. Read more information on sharing protocols at https://plos.org/protocols?utm_medium=editorial-email&utm_source=authorletters&utm_campaign=protocols.

We look forward to receiving your revised manuscript.

Kind regards,

Mohsan Ullah

Academic Editor

PLOS ONE

Journal Requirements:

4. Please note that funding information should not appear in any section or other areas of your manuscript. We will only publish funding information present in the Funding Statement section of the online submission form. Please remove any funding-related text from the manuscript.

 "This study was performed in the framework of the Chair for Avian Biosecurity and Health, hosted by the National Veterinary College of Toulouse and funded by the Direction Générale de l’Alimentation, Ministère de l’Agriculture et de l’Alimentation, France. Jean Artois and Simon Dellicour have received funding from the European Union’s Horizon 2020 research and innovation programme under grant agreement No 874850. The contents of this publication are the sole responsibility of the authors and don't necessarily reflect the views of the European Commission. Simon Dellicour also acknowledges support from the Fonds National de la Recherche Scientifique (F.R.S.-FNRS, Belgium; grant n°F.4515.22) and from the Research Foundation - Flanders (Fonds voor Wetenschappelijk Onderzoek-Vlaanderen, FWO, Belgium; grant n°G098321N)."

6. We note that you have indicated that there are restrictions to data sharing for this study. PLOS only allows data to be available upon request if there are legal or ethical restrictions on sharing data publicly. For more information on unacceptable data access restrictions, please see http://journals.plos.org/plosone/s/data-availability#loc-unacceptable-data-access-restrictions. 

7. We note that Figures 1-3 and Supplementary figures 1-3 in your submission contain map/satellite images which may be copyrighted. All PLOS content is published under the Creative Commons Attribution License (CC BY 4.0), which means that the manuscript, images, and Supporting Information files will be freely available online, and any third party is permitted to access, download, copy, distribute, and use these materials in any way, even commercially, with proper attribution. For these reasons, we cannot publish previously copyrighted maps or satellite images created using proprietary data, such as Google software (Google Maps, Street View, and Earth). For more information, see our copyright guidelines: http://journals.plos.org/plosone/s/licenses-and-copyright.

a. You may seek permission from the original copyright holder of Figures 1-3 and Supplementary figures 1-3 to publish the content specifically under the CC BY 4.0 license.  

Reviewers' comments:

Reviewer's Responses to Questions

**Comments to the Author**

1. Is the manuscript technically sound, and do the data support the conclusions?

Reviewer #1: Yes

Reviewer #2: Yes

2. Has the statistical analysis been performed appropriately and rigorously? 

Reviewer #1: Yes

Reviewer #2: Yes

3. Have the authors made all data underlying the findings in their manuscript fully available?

Reviewer #1: Yes

Reviewer #2: Yes

4. Is the manuscript presented in an intelligible fashion and written in standard English?

Reviewer #1: Yes

Reviewer #2: Yes

5. Review Comments to the Author

Reviewer #1: In summary, the research provides valuable insights into the spatial risk factors and predictive models for HPAI infection in France.

The study's focus on accessible data and high-risk areas makes it highly relevant for practical applications in disease control.

Recommendations for future research could include exploring additional predictor variables or applying the model to other regions or types of poultry.

This analysis suggests that the research is well-positioned to contribute to both scientific understanding and practical efforts to manage HPAI, with some room for improvement in the areas of methodological transparency and supplementary materials.

For a better understanding of this phenomenon, and also to ensure a complete and objective overview, I recommend that you cite a scientific article:

Iancu, I. Tirziu, E. Pascu, C., Costinar, L., Degi, J., Badea, C., Gligor, A., Bucur, I., Popa, SA., Herman, V. EVOLUTION OF HPAI AVIAN INFLUENZA VIRUS STRAINS IN EUROPE BETWEEN 2005 AND 2023, Rev Rom Med Vet (2024) 34 | 1: 138-144, WOS:001253398200004.

https://agmv.ro/wp-content/uploads/2024/03/138_144_Iancu.pdf

Nicoleta Foltoș (Negrilă), G. Brănescu, V.B. Vuță, S. Bărăităreanu, Florica Bărbuceanu The circulation of highly pathogenic avian influenza viruses (HPAIV) among non-human mammals worldwide during the period 01.01.2019 – 30.04.2024 Rev Rom Med Vet (2024) 34 | 2: 27-35.

https://agmv.ro/wp-content/uploads/2024/06/27_35_Foltos_9_C_4_11zon.pdf

Reviewer #2: I am happy to see the elaboration of the manuscript. But, it has some scientific and grammatical issues in the current version, and I have highlighted some issues here in the attached MS file. Therefore, the present draft needs minor revision before further process.

6. PLOS authors have the option to publish the peer review history of their article (what does this mean?). If published, this will include your full peer review and any attached files.

Reviewer #1: No

Reviewer #2: **Yes: **Muhammad Rizwan

---

## [Author Response · Author response to Decision Letter 0]

22 Oct 2024

Journal Requirements:

 The manuscript has been updated according to the journal requirements

The following sentence has been added to the Methods: “Additional information regarding the ethical, cultural, and scientific considerations specific to inclusivity in global research is included in the Supporting Information (S1 Checklist)”

The questionnaire has been added to the Supporting Information.

 The code has been made available on https://zenodo.org/records/13765228 (DOI 10.5281/zenodo.13765227). The following sentence has been added tot the Methods: “The R code about the spatial smoothing of variables is available online (10.5281/zenodo.13765228).”

4. Please note that funding information should not appear in any section or other areas of your manuscript. We will only publish funding information present in the Funding Statement section of the online submission form. Please remove any funding-related text from the manuscript.

 The funding-related text has been removed from the manuscript.

 "This study was performed in the framework of the Chair for Avian Biosecurity and Health, hosted by the National Veterinary College of Toulouse and funded by the Direction Générale de l’Alimentation, Ministère de l’Agriculture et de l’Alimentation, France. Jean Artois and Simon Dellicour have received funding from the European Union’s Horizon 2020 research and innovation programme under grant agreement No 874850. The contents of this publication are the sole responsibility of the authors and don't necessarily reflect the views of the European Commission. Simon Dellicour also acknowledges support from the Fonds National de la Recherche Scientifique (F.R.S.-FNRS, Belgium; grant n°F.4515.22) and from the Research Foundation - Flanders (Fonds voor Wetenschappelijk Onderzoek-Vlaanderen, FWO, Belgium; grant n°G098321N)."

The following sentence has been added to the funding statement: “The funders had no role in study design, data collection and analysis, decision to publish, or preparation of the manuscript.”

6. We note that you have indicated that there are restrictions to data sharing for this study. PLOS only allows data to be available upon request if there are legal or ethical restrictions on sharing data publicly. For more information on unacceptable data access restrictions, please see http://journals.plos.org/plosone/s/data-availability#loc-unacceptable-data-access-restrictions. 

Information about data sharing has been clarified in the Data Availability Statement: “Detailed data about the poultry farm densities and poultry trade used in this study are not publicly accessible due to sensitive farmer information and are owned by the Direction Générale de l’Alimentation, Ministère de l’Agriculture et de l’Alimentation (DGAl) of the Ministère de l’Agriculture et de la Souveraineté Alimentaire, France. Data requests can be made to the DGAl. Data about poultry outbreaks can be found on the Global Animal Disease Information System (https://empres-i.apps.fao.org) of the Food and Agriculture Organization of the United Nations (FAO). Shapefiles used to create maps can be obtained from https://www.data.gouv.fr/fr/datasets/admin-express/ under the Etalab Open License (equivalent to CC-BY 2.0) from the Institut National de l’Informaiton Géographique et Forestiere (IGN). The Global Administrative Unit Layers (GAUL) dataset was also used (Urbano, Ferdinando (2018): Global administrative boundaries. European Commission, Joint Research Centre (JRC) [Dataset] PID: http://data.europa.eu/89h/jrc-10112-10004). The code has been made available on https://zenodo.org/records/13765228 (DOI 10.5281/zenodo.13765227).

7. We note that Figures 1-3 and Supplementary figures 1-3 in your submission contain map/satellite images which may be copyrighted. All PLOS content is published under the Creative Commons Attribution License (CC BY 4.0), which means that the manuscript, images, and Supporting Information files will be freely available online, and any third party is permitted to access, download, copy, distribute, and use these materials in any way, even commercially, with proper attribution. For these reasons, we cannot publish previously copyrighted maps or satellite images created using proprietary data, such as Google software (Google Maps, Street View, and Earth). For more information, see our copyright guidelines: http://journals.plos.org/plosone/s/licenses-and-copyright.

a. You may seek permission from the original copyright holder of Figures 1-3 and Supplementary figures 1-3 to publish the content specifically under the CC BY 4.0 license.  

We are not using copyrighted map figures. Shapefiles used to create maps were obtained from https://www.data.gouv.fr/fr/datasets/admin-express/ under the Etalab Open Licence (https://www.etalab.gouv.fr/wp-content/uploads/2014/05/Licence_Ouverte.pdf) (equivalent to CC-BY 2.0) from the Institut National de l’Informaiton Géographique et Forestiere (IGN). We are also using the Global Administrative Unit Layers (GAUL) dataset (Urbano, Ferdinando (2018): Global administrative boundaries. European Commission, Joint Research Centre (JRC) [Dataset] PID: http://data.europa.eu/89h/jrc-10112-10004). The following sentence has been added to each figure caption: “Shapefiles used to create maps are based on administrative boundaries available in the public domain.”

The captions have been added at the end of the manuscript:

“Supporting Information 

S1 Fig. Smoothing of the spatial predictors. Spatial representation of a given municipality and neighboring municipalities across four spatial scales, from the level 1 (olive green) to the level 4 (dark blue). Shapefiles used to create maps are based on administrative boundaries available in the public domain.

S1 File. Spatial distribution of predictor variables. Predictor variables included the number of poultry (chicken and duck) houses, the density of chicken houses per km2, the density of duck houses per km2, the density of fattening duck (all stages) houses per km2, the density of fattening duck (breeding stage) houses per km2 and the density of fattening duck (force-feeding stage) houses per km2. Shapefiles used to create maps are based on administrative boundaries available in the public domain.

S2 Fig. Comparison of high-risk areas obtained using different statistical models. The final model results are represented on top of the figure and the alternative model results are in the bottom part of the figure. The posterior means of the proportion of HPAI H5N8 infected poultry houses per municipality are represented on the left panels and the probability of having at least one HPAI H5N8 poultry outbreak per commune are represented on the right panels. Shapefiles used to create maps are based on administrative boundaries available in the public domain.

S1 Table. Summary of the final binomial iid models fits.

The reference list has been updated.

Reviewers' comments:

Reviewer's Responses to Questions

Comments to the Author

1. Is the manuscript technically sound, and do the data support the conclusions?

Reviewer #1: Yes

Reviewer #2: Yes

2. Has the statistical analysis been performed appropriately and rigorously?

Reviewer #1: Yes

Reviewer #2: Yes

3. Have the authors made all data underlying the findings in their manuscript fully available?

Reviewer #1: Yes

Reviewer #2: Yes

4. Is the manuscript presented in an intelligible fashion and written in standard English?

Reviewer #1: Yes

Reviewer #2: Yes

5. Review Comments to the Author

Reviewer #1: In summary, the research provides valuable insights into the spatial risk factors and predictive models for HPAI infection in France.

The study's focus on accessible data and high-risk areas makes it highly relevant for practical applications in disease control.

Recommendations for future research could include exploring additional predictor variable

---

## [Decision Letter · Decision Letter 1]

9 Dec 2024

Spatial risk modelling of highly pathogenic avian influenza in France: fattening duck farm activity matters

PONE-D-24-25856R1

Dear Dr. Claire Guinat,

We’re pleased to inform you that your manuscript has been judged scientifically suitable for publication and will be formally accepted for publication once it meets all outstanding technical requirements.

Kind regards,

Mohsan Ullah

Academic Editor

PLOS ONE

Additional Editor Comments (optional):

Reviewers' comments:

Reviewer's Responses to Questions

**Comments to the Author**

1. If the authors have adequately addressed your comments raised in a previous round of review and you feel that this manuscript is now acceptable for publication, you may indicate that here to bypass the “Comments to the Author” section, enter your conflict of interest statement in the “Confidential to Editor” section, and submit your "Accept" recommendation.

Reviewer #1: All comments have been addressed

Reviewer #3: All comments have been addressed

2. Is the manuscript technically sound, and do the data support the conclusions?

Reviewer #1: Yes

Reviewer #3: Yes

3. Has the statistical analysis been performed appropriately and rigorously? 

Reviewer #1: Yes

Reviewer #3: Yes

4. Have the authors made all data underlying the findings in their manuscript fully available?

Reviewer #1: Yes

Reviewer #3: Yes

5. Is the manuscript presented in an intelligible fashion and written in standard English?

Reviewer #1: Yes

Reviewer #3: Yes

6. Review Comments to the Author

Reviewer #1: After reevaluating the article, we consider that the suggestions have been taken into account and the changes have been made.

Reviewer #3: I would like to express my positive evaluation of the manuscript. After carefully reviewing the content, I find the study to be well-conducted and highly relevant to the field. The authors have thoroughly addressed the key issues raised during the review process, and the revisions have significantly enhanced the clarity and quality of the work. The methodology is sound, and the conclusions are well-supported by the data presented. I believe this manuscript makes a valuable contribution to the literature, and I recommend it for acceptance without further revisions.

7. PLOS authors have the option to publish the peer review history of their article (what does this mean?). If published, this will include your full peer review and any attached files.

Reviewer #1: No

Reviewer #3: No

---

## [Editor Report · Acceptance letter]

20 Dec 2024

PONE-D-24-25856R1 

PLOS ONE

Dear Dr. Guinat, 

I'm pleased to inform you that your manuscript has been deemed suitable for publication in PLOS ONE. Congratulations! Your manuscript is now being handed over to our production team.

Kind regards, 

on behalf of

Dr. Mohsan Ullah 

Academic Editor

PLOS ONE